# Age and vaccine information sources drive vaccine hesitancy: A household survey in Central-Western Brazil

Ana Isabel do Nascimento[1], Danilo dos Santos Conrado[1], Lisany Krug Mareto[1], Micael Viana de Azevedo[2], João Cesar Pereira da Cunha[2], Gabriel Serrano Ramires Koch[2], Laysa Gomes Osório[2], Samara Tessari Pires[2], Letícia Suemi Arakaki[2], Sara Raquel Pinto Borges[2], Robson Franca Gomes e Silva[2], Rodrigo Mayer Pucci[2], João Guilherme de Novaes Corrêa[2], João Vitor Barrio[2], Maria Eduarda de Souza Rodrigues[2], Artur Jorge Bianchi[2], Márcio José de Medeiros[3], Ana Paula Sayuri Sato[4], Maria Elizabeth Araújo Ajalla[2], Cláudia Du Bocage Santos-Pinto[2], Everton Falcão de Oliveira[1,2]*

1 Programa de Pós-Graduação em Doenças Infecciosas e Parasitárias, Universidade Federal de Mato Grosso do Sul, Campo Grande, MS, Brasil, 2 Faculdade de Medicina, Universidade Federal de Mato Grosso do Sul, Campo Grande, MS, Brasil, 3 Instituto Politécnico, Universidade Federal do Rio de Janeiro, Macaé, RJ, Brasil, 4 Faculdade de Saúde Pública, Universidade de São Paulo, São Paulo, SP, Brasil

* everton.falcao@ufms.br

## Abstract

In recent decades, the decline in vaccination coverage has garnered global attention, and the impact of vaccine hesitancy has become a significant concern for public health policymakers worldwide. This study aims to measure vaccine hesitancy and its associated factors among residents of Campo Grande, Mato Grosso do Sul, Brazil. From September 2022 to October 2023, a cross-sectional study aligned with a household survey was conducted to measure vaccination coverage among residents of Campo Grande municipality in Brazil. Data were collected through face-to-face interviews using the WHO. Reasons for vaccine hesitancy were classified under the 3C conceptual model of vaccine hesitancy determinants. Descriptive statistics were employed to characterize the study population, and univariate and multivariate logistic regression analyses were conducted to assess the association between hesitant and non-hesitant participants and the study variables. We included 467 households in the study, with a total of 518 participants interviewed. Over half of the participants hesitated to get vaccinated (50.2%), with the COVID-19 vaccines being the most hesitated (55.4%). The majority of reported reasons for hesitancy were related to a lack of confidence (62.3%). The hesitant participants in our study were younger than the non-hesitant participants (aOR = 0.98; 95% CI: 0.97, 0.99), were less likely to believe that vaccines could protect themselves and their children from serious diseases (aOR = 0.23; 95% CI: 0.06, 0.66), and were less likely to get information from healthcare workers or official health organizations' online profiles (aOR = 0.39; 95% CI: 0.17, 0.86). We observed a high prevalence of vaccine hesitancy in Campo Grande. The

**Data availability statement:** All relevant data are available within the manuscript and its supplementary files. The analyzed sample is relatively small, and therefore, even de-identified data could potentially allow the identification of individual participants. For this reason, the dataset cannot be made publicly available. Access to the dataset may be granted upon justified request and subject to approval by the Research Ethics Committee of the Federal University of Mato Grosso do Sul (UFMS), which is the institutional body responsible for oversight of ethical use and protection of participant information. Third-party institutional contact: Research Ethics Committee – Federal University of Mato Grosso do Sul (UFMS) Cidade Universitária, Campo Grande – MS, Brazil, Email: cepconep.propp@ufms.br.

**Funding:** This study was financed in part by the Coordenação de Aperfeiçoamento de Pessoal de Nível Superior, Brasil (CAPES, https://www.gov.br/capes/pt-br; Finance Code 001) (EFO) and by Universidade Federal de Mato Grosso do Sul, Brazil (UFMS, https://www.ufms.br/) (EFO). The funders had no role in study design, data collection and analysis, decision to publish, or preparation of the manuscript. The funders had no role in study design, data collection and analysis, decision to publish, or preparation of the manuscript.

**Competing interests:** I have read the journal's policy and the authors of this manuscript have the following competing interests: EFO is an Academic Editor of PLOS ONE. This does not alter our adherence to PLOS ONE policies on sharing data and materials.

results highlight the potential impact of the COVID-19 pandemic and the infodemic in increasing negative feelings about vaccines.

## Introduction

Vaccination has been recognized as one of the best methods for controlling the transmission of infectious diseases [1]. In Brazil, the Brazilian National Immunization Program (*Programa Nacional de Imunizações*, PNI), was established as the main institutional strategy for implementing vaccination as a public health policy, building on earlier successful initiatives such as the variola eradication campaign, which aimed to control and eradicate smallpox [2]. This achievement occurred in an adverse national context marked by political instability, limited resources for preventive health measures, and profound social and demographic changes, including rapid urbanization driven by internal migration and persistent sanitation challenges [2]. In response, the Brazilian National Immunization Program (PNI) was formulated in 1973 and formally established in 1975, following the success of the smallpox eradication campaign, to ensure sustained vaccination capacity and strengthen the foundations of the Brazilian Unified Health System, one of the largest public health systems in the world [2,3].

Since its establishment, the PNI has ensured nationwide access to vaccines free of charge, achieving substantial operational and structural advances [3]. These efforts have led to the control of multiple vaccine-preventable diseases, including poliomyelitis, tetanus, and urban yellow fever, and have contributed significantly to reductions in infant mortality across the country [4]. The PNI also developed widely recognized communication strategies, such as partnerships with media figures, mass radio and television campaigns, and the creation of the nationally recognized character *Zé Gotinha*, which played a key role in the successful control of poliomyelitis [3].

Nevertheless, vaccination coverage has declined globally in recent years [5]. In Brazil, despite historically high coverage levels, a sustained downward trend has been observed since 2016 [6]. This decline has coincided with the growing influence of anti-vaccine discourse, the politicization of vaccination during the COVID-19 pandemic, and the widespread circulation of misinformation in digital environments, factors that may have contributed to increasing mistrust and changes in public perceptions of immunization.

Understanding the reasons behind the choice to vaccinate or not – and interpreting these reasons regarding the context and the perceptions about vaccines and illnesses, considering sociodemographic factors – is crucial for sustaining high vaccine demand [7]. This concern led to the development of the concept of "vaccination hesitancy", a phenomenon declared as one of the global threats to public health in 2019 and broadened the theoretical framework surrounding vaccination attitudes [8]. This concept highlights the diversity within the vaccine-hesitant population, which spans a spectrum from complete acceptance to outright refusal of vaccines. The World Health

Organization (WHO) defines, in 2015, vaccination hesitancy as the "delay in acceptance or refusal of vaccines despite the availability of vaccination services" [9,10].

Vaccine hesitancy fundamentally stems from a decision-making process influenced by a variety of factors, summarized in the 3C model of vaccine hesitancy: complacency, confidence, and convenience. Complacency refers to the underestimation of disease risk and the perceived need for vaccination. Confidence-related determinants encompass mistrust in the vaccine, the health system distributing it, its professionals, and the policymakers governing vaccine policies. Lastly, determinants related to convenience involve issues of geographical access, availability, comprehensibility of vaccine-related information, quality of health services, and other access-related obstacles [10]. Subsequently, the 5C conceptual model was developed to incorporate additional behavioral aspects of vaccination, such as collective responsibility and risk calculation [11], alongside the renewal of the conceptual model, the WHO redefined vaccination hesitancy as "a motivational state of being conflicted about or opposed to, getting vaccinated", which included intentions and willingness [12].

Following the global trend of declining vaccination coverage rates [6], Campo Grande, the capital of Mato Grosso do Sul, located in a tri-border region (Brazil-Bolivia-Paraguay), has experienced a decline in vaccination coverage since 2019 [13]. Based on this context, this study aimed to measure vaccination hesitancy and identify its associated factors among residents of Campo Grande.

## Materials and methods

### Study design and period

This cross-sectional study used data collected through a household survey conducted in Campo Grande, Mato Grosso do Sul, Brazil, between September 2022 and October 2023, with the primary objective of estimating vaccination coverage in the municipality. For the purposes of this analysis, data on vaccine hesitancy were also collected and analysed.

### Study site

Campo Grande, the capital state of Mato Grosso do Sul, by the time of our study, comprised a population of 898,100 inhabitants. The municipality holds a population mostly female and young (sex ratio = 92.2; age median = 34 years) [14]. Mato Grosso do Sul shares borders with Bolivia and Paraguay, holding, alongside Mato Grosso, over 60% of the immigrant population of the Brazilian Central-West region between 2022 and 2023 [15]. In the same period, Mato Grosso do Sul provided the second-highest amount of refugee concessions in the region. The state received four times the migrant population of 2013 [15]. In 2022, 5,614 migrant workers were received in the municipality [15]. Therefore, healthcare actions to promote vaccination are common in the municipality of Campo Grande, targeting immigrants and refugees.

### Sampling

The sampling method was cluster sampling, according to the one proposed by the WHO, in 2018, for studies that aim to estimate vaccination coverage [16]. Further details of the sampling method and calculation are described in the S1 Appendix. This method is based on two stages: (1) cluster selection and (2) households' selection.

(1) Definition and selection of clusters: Assuming that the expected average vaccination coverage in Campo Grande, for all vaccines available in the PNI, is 90%, with a confidence interval around the estimates of 8% (i.e., 90% ± 8% coverage estimate), with an alpha (type I error) of 5%, the effective sample size – based on an assumption of simple random sampling – was $n = 101$. To determine the average number of people eligible for the study (individuals aged 12 years or older), a pilot study was conducted. For this, one cluster (census tract) was randomly selected. Considering that on average, in Brazil, each census tract has approximately 300 households, for the pilot study 10% of the households contained in the selected cluster were drawn, which resulted in 31 households. After completing the pilot study, the average number of respondents per cluster, within a 3-hour interval, with a field team of 6 researchers, distributed

in pairs, was 10. Assuming that the intracluster correlation is 0.33 [16], the design effect size was set at 3. Applying the formula proposed by the WHO, the estimated number of clusters was 30.3, which was rounded to 30. The selection of clusters was done by simple random sampling without replacement, using the cartographic base of census sectors from the IBGE of 2021. The cluster where the pilot study was conducted was included in the study. Therefore, an additional 29 clusters were sampled afterward. Clusters that primarily contained institutionalized populations (prisons and long-term care facilities, such as nursing homes) were immediately replaced when drawn. Clusters containing large condominiums or gated communities that did not allow the study team entry for data collection after initial contact by the researchers were also replaced.

(2) The definition of the number and selection of the households was based on the pilot study. The number of residencies visited to find an eligible participant averaged 1.5, the inflation factor to account for refusals and non-respondent residences was 1.05, and the average number of respondents per day of data collection was 10. Therefore, the number of households per cluster was defined as 15. Those households were selected by random simple probabilistic sampling.

The random sampling and spatial allocation of clusters and households were performed using *sf* package from software R 3.4.2.

## Study population and data collection

All residents of the Campo Grande municipality, aged 12 years or more, who consented to participate in this study, were eligible. The study was based on data collected through an interview using the SAGE Work Group questionnaire from 2015, which was translated into Portuguese and linguistically adapted to fit our context, without deviating from its original meaning [9,10,17]. Questions about socioeconomic and demographic variables, along with access to health units with vaccination facilities, were included in this interview and data collection instrument. During the interviews, participants were informed about the concept of vaccination hesitancy, proposed by the WHO in 2015, since this study's instrument was developed based on that definition [10], and answered about the vaccine hesitancy related to their own vaccination and their reasons for the hesitation. Finally, we classified the reasons for hesitancy under the 3C conceptual model for vaccine hesitancy, as the data collection questionnaire that we used was built based on this model (S3 Appendix). When possible, these reasons were also interpreted in light of the 'risk calculation' dimension (the individual assessment of the risks and benefits of vaccination) from the 5C model of vaccine hesitancy [11].

## Statistical analysis

Descriptive statistics was employed to characterize the studied population. The study data were analysed according to the occurrence of vaccine hesitancy. Therefore, the study population was divided into hesitant (HP) and non-hesitant (NHP) participants. The continuous variables were reported as mean values and standard deviation (SD) and were compared with the Welch *t*-test. The categorical variables were reported as a frequency and were compared with the chi-square and/ or Fisher's exact tests in this stage.

We conducted a multivariate logistic regression model to assess the relationship between vaccine hesitancy and the covariates found to have a p-value of 0.20 or less were included in the previous stage of the analysis. The stepwise algorithm (considering both backward and forward directions) and the Akaike Information Criterion (AIC) were used for variable selection, controlling for potential confounders, and determining the best-fitting model. The presence of multicollinearity was assessed using the variance inflation factor (VIF). The Hosmer-Lemeshow test was used as a measure of fit.

To explore heterogeneity across age groups, we conducted additional analyses using categorized age. Age was grouped into 20-year intervals to ensure sufficient sample size within each category. Descriptive analyses compared the distribution of vaccine hesitancy across age groups, and a multivariable logistic regression model was re-estimated using age as a categorical variable, applying the same modelling strategy as in the primary analysis (see S2 Appendix). This

complementary approach was adopted to facilitate interpretation of potential generational patterns, while age was retained as a continuous variable in the main model to preserve statistical power.

The significance level adopted for all hypothesis tests was 5% ($\alpha = 0.05$). The analysis was performed using R software version 4.3.2 (https://www.r-project.org/), and the following packages were used: *tidyverse*, *descr*, and *generalhoslem*.

### Ethics approval and consent to participate

The study was approved by the Research Ethics Committee (CEP) of the Federal University of Mato Grosso do Sul (CAEE: 47947821.0.0000.0021), according to opinion nº: 5.200.726. All the participants signed a written informed consent form before the data collection. When adults the Informed Consent Form (ICF) was applied, and the Assent Form (AF) was applied for minors (under 18 years old), along with the consent of the responsible guardian present during the application of the questionnaire, who also signed the ICF.

### Results

A total of 467 households were included and 518 participants were interviewed. The medium of respondents was 1,1 (SD = 0.38) per household, and 17 respondents per cluster (SD = 4).

Most of the participants were female (321/518; 62.0%), and non-white (362/518; 69.9%). The mean age of the population was 46,7 years, most participants were between 52–71 (179/518; 34.6%), and 32–51 years old (174/518; 33.6%). The mean of years of study was 9.7. Most of the respondents were not classified as low income (334/518; 64,5%), and the resided households had an average of 3.1 residents per household. Most participants lived in houses with access to piped water (494/518; 95.4%), and sanitary sewer treatment (273/518; 52.7%), but the majority didn't have health insurance (363/518; 70.1%). The socioeconomics and demographic characteristics of the participants of the study are presented in Table 1.

Regarding access to health units with vaccination facilities and the availability of vaccines in these locations, 23.0% (119/518) of the respondents reported barriers. Most of the mentioned reasons were related to the lack of vaccines (63/119; 52.9%) and the waiting time at the health unit (50/119; 42.0%). Other reasons for not getting vaccinated in the recommended schedule were reported by 6.2% (32/518) of the participants, as medical counterindication to get vaccinated due to illness or allergies, lack of professionals, lack of support to get to the health unit, bad reception in the health unit, and refusal of the professionals to vaccinate them. A large percentage of respondents affirmed to have had received guidance from health professionals about vaccination (392/518; 75.8%). Regarding the relationship with health professionals, most of the participants had a great or good relationship with the health professionals and workers from the attended health unit (367/518; 69.7%).

Over half of the respondents (71.8%, 372/518) said to believe that most people who live with them have themselves vaccinated with all the recommended vaccines. And 77.6% (402/518) reported to not have changed their beliefs after the beginning of the pandemic of COVID-19. However, the HP had a greater percentage of respondents who believed that most people who live with them do not get vaccinated as recommended by the PNI (28.9%, 75/260) when compared to the NHP (19.8, 51/258) with difference between groups (p = 0.020).

The majority of respondents did not believe that there are reasons for not vaccinating people in the community (87.3%. 452/518). Nonetheless, the HP had a greater percentage of participants who did believe in reasons for not vaccinating the population (16.2%, 42/260) when compared to the NHP (9.3%, 24/258) (p = 0.027). A greater percentage of the HP also reported to have developed this belief after the onset of the pandemic of COVID-19 (50.0%, 21/42) (p = 0.035). The reasons reported by the NHP for the non-vaccination of the community were mainly related to medical counterindication and illnesses (66.7%, 16/24) whilst, among the HP, the reasons described were the lack of safety in the vaccination and the short time for its development (35.7%, 15/42) and the freedom for the individual choice (28.6%, 12/42).

**Table 1. Study data according to the occurrence of vaccine hesitancy.**

| | Total (*n*= 518) | NHP (*n* = 258) | HP (*n* = 260) | OR [95% CI] | *p*-value |
|---|---|---|---|---|---|
| Age (mean, (DP)) | 46.7 (17.4) | 49.6 (18.0) | 43.8 (16.3) | 0.98 [0.97; 0.99] | <0.001 |
| Years of study (mean, (DP)) | 9.7 (4.6) | 9.5 (4.6) | 9,9 (4.6) | 1.02 [0.98;1.06] | 0.297 |
| Residents per household (mean, (DP)) | 3.1 (1.5) | 3.0 (1.5) | 3,1 (1.5) | 1.06 [0.94;1.19] | 0.304 |
| Sex | | *n* (%) | *n* (%) | | |
| Female | 321 (62.0%) | 158 (61.2%) | 163 (62.7%) | Reference | 0.803 |
| Male | 197 (38.0%) | 100 (38.8%) | 97 (37.3%) | 0.94 [0.66;1.34] | |
| Ethnicity | | | | | |
| White | 156 (30.1%) | 74 (28.7%) | 82 (31.5%) | Reference | 0.540 |
| Non-white | 362 (69.9%) | 184 (71.3%) | 178 (68.5%) | 0.87 [0.60;1.27] | |
| Low income | | | | | |
| Yes | 184 (35.5%) | 88 (34.1%) | 96 (36.9%) | Reference | 0.564 |
| No | 334 (64.5%) | 170 (65.9%) | 164 (63.1%) | 0.88 [0.62;1.27] | |
| Access to piped water | | | | | |
| Yes | 494 (95.4%) | 248 (96.1%) | 246 (94.6%) | Reference | 0.543 |
| No | 24 (4.6%) | 10 (3.9%) | 14 (5.38%) | 1.40 [0.61;3.35] | |
| Access to sanitary sewer treatment | | | | | |
| Yes | 273 (52.7%) | 140 (54.3%) | 133 (51.2%) | Reference | 0.535 |
| No | 245 (47.3%) | 118 (45.7%) | 127 (48.8%) | 1.13 [0.80;1.60] | |
| Access to health insurance | | | | | |
| Yes | 155 (29.9%) | 86 (33.3%) | 69 (26.5%) | Reference | 0.111 |
| No | 363 (70.1%) | 172 (66.7%) | 191 (73.5%) | 1.38 [0.95;2.02] | |
| Do you believe that vaccines can protect yourself and children from serious diseases? | | | | | |
| Yes | 496 (95.9%) | 253 (98.4%) | 243 (93.5%) | Reference | 0.008 |
| No | 21 (4.1%) | 4 (1.56%) | 17 (6.54%) | 4.29 [1.55;15.5] | |
| Do you believe there are reasons for people not get vaccinated? | | | | | |
| Yes | 66 (12.7%) | 24 (9.30%) | 42 (16.2%) | Reference | 0.027 |
| No | 452 (87.3%) | 234 (90.7%) | 218 (83.8%) | 0.53 [0.31;0.91] | |
| Has distance, opening hours of the health unit, time needed to get to the health unit, wait at health unit prevented you from getting yourself immunized? | | | | | |
| Yes | 119 (23.0%) | 52 (20.2%) | 67 (25.8%) | Reference | 0.157 |
| No | 399 (77.0%) | 206 (79.8%) | 193 (74.2%) | 0.73 [0.48;1.10] | |
| Have you ever received or heard negative information about vaccination? | | | | | |
| Yes | 391 (75.8%) | 194 (75.5%) | 197 (76.1%) | Reference | 0.960 |
| No | 125 (24.2%) | 63 (24.5%) | 62 (23.9%) | 0.97 [0.65;1.45] | |
| What source of information you mostly use to know about vaccination related information? | | | | | |
| Health care worker/websites or profile of official Health Organizations | 130 (25.1%) | 79 (30.6%) | 51 (19.6%) | Reference | <0.001 |
| Traditional News (official website or television) | 232 (44.8%) | 126 (48.8%) | 106 (40.8%) | 1.30 [0.84;2.02] | |
| Social media | 120 (23.2%) | 41 (15.9%) | 79 (30.4%) | 2.97 [1.78;5.01] | |
| Others (friends, neighbours, or none) | 36 (7.0%) | 12 (4.7%) | 24 (9.2%) | 3.06 [1.42;6.89] | |
| Have you ever been advised by health professionals about vaccination? | | | | | |
| Yes | 392 (75.8%) | 195 (75.6%) | 197 (76.1%) | Reference | 0.977 |
| No | 117 (22.6%) | 59 (22.9%) | 58 (22.4%) | 0.97 [0.64;1.47] | |

*(Continued)*

**Table 1.** (Continued)

| | Total (*n*= 518) | NHP (*n* = 258) | HP (*n* = 260) | OR [95% CI] | *p*-value |
|---|---|---|---|---|---|
| Do not know/do not remember | 8 (1.6%) | 4 (1.6%) | 4 (1.5%) | 0.99 [0.22;4.45] | |
| What is your relationship between yourself and the health care workers from the Health Unit you attend? | | | | | |
| Great/Good | 361 (69.7%) | 187 (72.5%) | 174 (66.9%) | Reference | 0.314 |
| Reasonable/Indifferent | 76 (14.7%) | 38 (14.7%) | 38 (14.6%) | 1.07 [0.65;1.77] | |
| Bad | 30 (5.8%) | 11 (4.5%) | 19 (7.3%) | 1.84 [0.86;4.14] | |
| Do not attend any health care Unit | 51 (9.9%) | 22 (8.5%) | 29 (11.2%) | 1.41 [0.78;2.58] | |

As to other aspects related to the community's vaccination, almost half the participants did not believe there were difficulties for ethnical or religious groups in the community to get vaccinated (49.6%, 257/518). Of the 28.2% (146/518) that reported difficulties from those groups to get vaccinated, the main reason described was the self-choice to not get vaccinated (62.3%, 104/167), and 21.6% of those reasons were related to the lack of active research for those communities by the health units (36/167.). The great majority of participants reported having never experienced any community leader discouraging vaccination (81.1%, 420/518). The most reported leaders who discouraged vaccination were religious leaders (54.6%, 30/55) and politicians (20.0%, 11/55).

Among the study participants, 50,2% (260/518) reported to have hesitated to get one or more vaccines. Among the hesitant respondents, 62.9% (127/260) hesitated to vaccinate only after the beginning of the COVID-19 pandemic. In total, 19.7% (102/518) have refused to get vaccinated, and 56.9% (58/102) of the respondents who refused to get vaccinated with one or more vaccines have refused after the beginning of the COVID-19 pandemic.

The most hesitated vaccine were the COVID-19 vaccines (144/260; 55.4%), followed by the influenza vaccine (48/260; 18.5%). Other mentioned vaccines were DTP or DTP (27/260; 10.38%), Hepatitis B (12/260; 4.61%), meningitis (9/260; 3.46%), yellow fever (8/260; 3.07%), MMR (7/260; 2.69%), BCG (5/260; 1.92%), pneumococcal (4/260; 1.53%), IPV or OPV (2/260; 0.76%) vaccines. Among the hesitant respondents, 87 participants (87/260; 33.5%) reported to have delayed their vaccination schedule or to have not been vaccinated for a long time. Most of them informed to not recall which immunizers were out of date, to no longer hold their immunization booklet, or to not remember where it was kept.

Regarding reasons for hesitancy, over half were related to lack of confidence (162/260; 62.3%), and all of those were reported to have occurred after the onset of the COVID-19 pandemic. The low perception of safety or worries related to side effects were the most prevalent reason for lack of confidence (92/162; 56.8%), followed by the low perception of efficacy of the vaccine (68/162; 42.0%) and to have read or listened to negative news about the vaccine in various types of media (57/162; 35.2%). Other reasons for hesitation were related to complacency (115/260; 42.7%). The perception that the vaccination was not necessary (72/115; 62.6%), forgetting the date of the vaccine shot (35/115; 30.4%), and demotivation or laziness (13/115; 11.3%) were the main complacency motives informed, but, among those, only the perception that the vaccine was not needed was reported mainly due to the pandemic, moreover, other less mentioned reasons, such as political (4/115; 3.5%) and religious reasons (4/115; 4.4%) were also reported mostly because of the pandemic. Regarding the lack of convenience (93/260; 35.8%), 50.5% (47/93) could not get vaccinated because of their work schedule, and 32.3% (30/93) reported difficulties in finding reliable information about the hesitated vaccines, which was mostly given to the pandemic (Fig 1).

When comparing hesitant participants (HP) to non-hesitant ones, the HP was younger than the NHP (*p*<0.001). When compared to the NHP, the hesitant participants believed less in the ability of vaccines to protect themselves and children from serious illnesses (*p*=0.008). Among those who believed that there were reasons for people to not get vaccinated, the HP was the majority (*p*=0.027). The HP and NHP got informed about vaccination mostly through television, however,

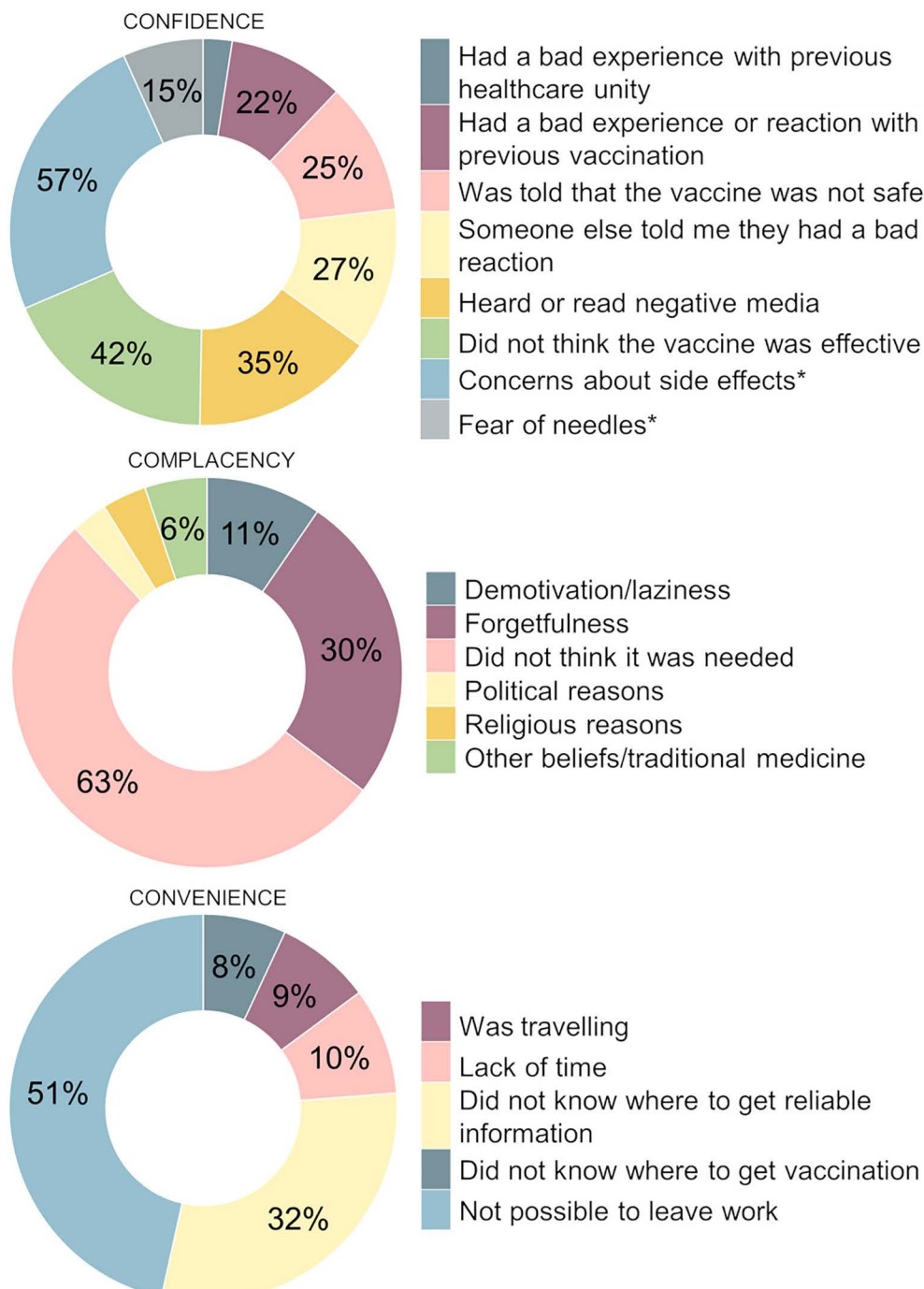

**Fig 1. Reasons for hesitation according to the 3C conceptual model of determinants of vaccination hesitancy.**

the HP also used social media as the main source of information or did not get actively informed about the topic, while the NHP got more informed through health care professionals or organizations ($p < 0.001$).

The covariates that remained in the final logistic regression model were: mean age, having access to health insurance, positive answer to "vaccines can protect you and children from serious illnesses", believing there are reasons for people not getting vaccinated, having difficulties accessing vaccines and vaccination services, getting informed mainly through health care professionals, or through social media, or through other sources (friends, family or not getting informed about vaccination at all).

Among these covariates, the mean age was significantly negatively associated with hesitancy. In the age-stratified analysis, individuals aged 52–71 showed lower levels of hesitancy (OR = 0.59; 95%CI = 0.36–0.95; $p = 0.031$) compared with other age groups (Table B in S2 Appendix). Believing that vaccines can protect oneself and children had a moderated negative association with hesitancy. Getting informed mainly through social media had a positive moderated association with hesitancy. The other variables showed no significant association with the increase in vaccine hesitancy, but they contributed to the fitting of the best model. The p-value for the Hosmer and Lemeshow test was 0.485, showing a good model adjustment (Table 2).

## Discussion

This study observed a high prevalence of vaccine hesitancy among participants, higher than that found in other studies conducted in the country and in countries with similar health systems before the COVID-19 pandemic [18,19]. This result highlights not only the possible impact of the pandemic on vaccination hesitancy in the country, compared to the pre-pandemic era, but also the higher hesitancy compared to other countries with similar universal health systems.

Regarding VH in Latin American countries, most studies took place in Brazilian municipalities. Most studies developed abroad, however, don't address general adult vaccination; instead, they investigate specific vaccine-hesitation, focusing on pregnant women and the elderly population. Nevertheless, COVID-19 vaccine hesitancy has been largely studied. In a scope review of COVID-19-specific hesitation in Africa and Latin America, the highest percentage of hesitancy was 67.2% in Egypt and 73.0% in Ecuador, higher than the percentage of hesitation observed in our study (27.8% of the total population). Anyhow, the lack of homogeneity in the methodology of studying this phenomenon in Latin American and African populations underscores the need for further understanding of the vaccination hesitancy in the global south.

The association between age and hesitancy was also identified in other studies [18,20]. This phenomenon is described in the literature as the paradox of immunization, experienced among younger individuals who exhibit more complacent behaviour due to a lack of memory of epidemics and pandemics in their lifetime [21,22]. This was further supported by

**Table 2. The final model for the occurrence of vaccine hesitancy.**

| Covariates | β Coefficient (SE) | aOR (95% CI) |
|---|---|---|
| Intercept | 2.68 (0.70) | 14.82 (2.04; 66.69) |
| Age | − 0.01** (0.00) | 0.98 (0.97; 0.99) |
| "Do you believe that vaccines can protect yourself and children from serious diseases?" (Yes) | − 1.46* (0.59) | 0.23 (0.06; 0.66) |
| Main source of information about vaccination (Health care worker/ websites or profile of official Health Organizations) | − 0.93* (0.40) | 0.39 (0.17; 0.86) |
| Main source of information about vaccination (social media) | 0.04 (0.41) | 1.03 (0.45; 2.30) |
| Main source of information about vaccination (Traditional media: online and television journals) | − 0.59 (0.39) | 0.55 (0.25; 1.17) |

Legend: SE: Standard Error; aOR: adjusted odds ratio; CI: confidence interval.

*p-value < 0.05; **p-value < 0.01.

our age-stratified analysis, in which participants aged 52–71 years showed lower hesitancy compared with younger individuals.

Other contextual determinants related to vaccination hesitancy, particularly regarding COVID-19 vaccination, were described in two systematic reviews, where factors such as being female, non-white, having lower income, and lower education levels were determinants of higher vaccination hesitancy [23,24]. This profile is similar to that found in our study. Although variables other than age were not related to vaccine hesitancy, the absence of association underscores the importance of exploratory and periodic studies on factors related to hesitancy. COVID-19 vaccine-specific hesitancy was higher in our study compared to other studies conducted in the country [25,26]. Our study collected data through face-to-face interviews, which may have contributed to the higher prevalence of COVID-19-specific vaccine hesitancy observed. During the pandemic, many studies used online data collection methods, potentially limiting the participation of lower-income individuals and leading to an under-representation of hesitancy rates in the country [27].

The distrust related to the COVID-19 vaccine may be linked to and exacerbated by the COVID-19 infodemic [28]. The reduction in social contact due to social distancing has increased global screen time, leading to increased searches for information about vaccines amid the growing production of misinformation and disinformation regarding COVID-19 vaccines on various social platforms [29,30]. Despite the continued general recognition of the importance of vaccination observed in our study, confidence and feelings of collective responsibility may have diminished due to the pandemic [31,32].

Therefore, the HP in our study may have been more influenced by negative information about vaccines than the NHP. This result may be due to the increasing amount of unverified and negative online information during the pandemic, which may have intensified negative feelings toward the vaccines among these respondents, as observed in other studies [33,34].

Regarding influenza vaccine-specific hesitancy, other studies across the country found similarly low hesitancy, especially among the elderly population, who are the main target of vaccination campaigns [35,36]. This may be due to the widespread diffusion of the National Immunization Program (PNI) influenza immunization campaigns targeting the population annually [37].

Other reasons related to complacency, as reported by the participants, may be linked to the paradoxical success of vaccination, as described previously and recently observed with COVID-19 booster doses [38]. Some authors also report that complacent determinants are driven by the neoliberal system widely spread across the globe [39,40]. Neoliberalism stands for the 70's renewing of Adam Smith's liberalism, which defended minimal government intervention and rolling over economic matters, to grow the market [41]. Individualism, free market solutions through privatization, decentralization, and deregulation are pillars of the neoliberal mindset, increasingly incorporated in the rhetorical arguments of vaccine reluctant or hesitant advocates [39].

Forgetting about the vaccination [42,43] and not recording specific hesitated vaccines are described here as hesitation-related behaviors. Forgetfulness may result from a prioritization process where certain tasks are deemed more important than others, leading to the neglect of less prioritized tasks [44,45]. Methods for recalling vaccination doses, especially for long intervals between doses in adult and adolescent vaccination schedules, may be useful for increasing vaccination rates [46].

In addition, the association between risk perception and vaccination behavior has been well established in the scientific literature. Risk perception is commonly described as comprising three dimensions: the perceived likelihood of harm, perceived individual vulnerability to the hazard, and perceived severity of its consequences. Accordingly, higher levels of risk perception are associated with greater vaccine uptake [47]. Based on this framework, we hypothesize that complacent adults' management of their own vaccination may be partly explained by lower levels of perceived risk within this population.

The negative view towards mandatory vaccination found among the participants has been previously described in the Brazilian population and in some European countries [48,49]. This perception must be addressed, especially considering that the political scenario in Brazil may have influenced public opinion about vaccination during the pandemic [50].

The perception that most people who lived with the HP (friends, family, and neighbours) did not get vaccinated with all recommended vaccines evokes discussions about vaccination as a social norm. As an intrinsically social process, individual vaccination can influence not only vaccination coverage but also the behavior of other individuals within social circles as a social norm [51]. Therefore, valuing patients and communities who get vaccinated may enhance vaccination uptake among relatives, friends, and neighbours, and strengthen resilience against anti-vaccine sentiments [51].

Despite the low prevalence of obstacles to vaccination, the lack of vaccines and other access difficulties have been described in previous literature in the Global South and in Brazil [52,53]. However, this issue appears to be more related to a shortage of vaccines during the pandemic, which was a global occurrence. During the pandemic, the expectation that Brazil would be an independent source to supply itself and other Latin American countries did not materialize, highlighting the country's lack of self-sufficiency in vaccine manufacturing during a public health emergency [54].

Regarding the good relationship with healthcare professionals among the participants in our study, healthcare professionals are the most important and reliable source of information about vaccination, and they play a crucial role in enhancing health literacy among the population [55]. In recent years, primary healthcare attention has been strengthened in Brazil, which may have improved the study population's perception of healthcare. Nevertheless, the strengthening of the patient-healthcare professional is crucial to decrease hesitancy in the studied population. The development of strategies to quickly respond to health emergencies that may decrease vaccine uptake, with honesty and transparency, to enhance popular trust in the health system [56].

The use of traditional mass media, such as television, in combination with educational interventions, has been shown to improve perceptions of vaccination. At the same time, the widespread influence of social media and other digital communication technologies underscores the need for these platforms to be strategically incorporated into vaccination promotion and advocacy within healthcare system [57].

Our findings should be interpreted within a broader context of risk perception, institutional trust, and health communication. The WHO SAGE Working Group has emphasized that vaccine hesitancy is shaped not only by individual beliefs but also by perceived risk, confidence in health authorities, and the credibility of information disseminated to the public [10] (WHO, 2014). International evidence supports this perspective, as a global survey across 67 countries demonstrated substantial variation in vaccine confidence, with institutional trust and political context playing a particularly important role in middle-income settings [58]. During the COVID-19 pandemic, the World Health Organization further highlighted the emergence of an "infodemic," in which the rapid spread of misinformation through social and traditional media challenged effective risk communication and public adherence to health recommendations [59]. Although our study was not designed to directly assess political determinants or exposure to misinformation, the observed associations between vaccine hesitancy, younger age, and reliance on non-institutional information sources are consistent with this broader literature, underscoring the importance of strengthening trustworthy health communication and institutional credibility.

Our study has some limitations, including a sample that may not accurately represent the city's true demographics. Specifically, there was an underrepresentation of teenagers and an overrepresentation of the elderly compared to the actual population of Campo Grande. Additionally, there were few participants per household, and access to wealthier residents was restricted due to entry denial into gated communities and apartment buildings. Nevertheless, few studies have comprehensively addressed vaccination hesitancy across all vaccines and included a wide age range during the transition towards the end of the COVID-19 pandemic. Furthermore, our study was conducted through face-to-face interviews, potentially biasing towards a higher representation of the lower-income population in Campo Grande.

## Conclusions

We observed a high prevalence of vaccine hesitancy in Campo Grande, likely exacerbated by the COVID-19 pandemic and infodemic, which have contributed to heightened negative perceptions of vaccines. While access barriers were noted

to a lesser extent, they remain significant for achieving adequate vaccination coverage. Additionally, complacency and forgetfulness regarding vaccination importance may also impact vaccination rates in the municipality. Addressing these challenges at both individual and community levels is crucial. Effective use of technology for communication and combating misinformation is essential. Furthermore, promoting health education to enhance individual autonomy and community resilience is key to countering current and future anti-vaccine movements.

## Supporting information

**S1 Appendix. Sampling strategy and sample size calculation using the WHO two-stage cluster sampling method.**
(PDF)

**S2 Appendix. Analysis of vaccine hesitancy by age groups (20-year intervals): descriptive comparisons and multivariable logistic regression.**
(PDF)

**S3 Appendix. Data collection instrument: adapted WHO SAGE questionnaire, additional variables, and 3C model framework for vaccine hesitancy.**
(PDF)

## Acknowledgments

We sincerely thank the residents of Campo Grande who graciously welcomed our study team and participated in this research. Your valuable insights and cooperation have been instrumental in advancing our understanding of vaccination hesitancy in the community.

## Author contributions

**Conceptualization:** Márcio José de Medeiros, Ana Paula Sayuri Sato, Maria Elizabeth Araújo Ajalla, Cláudia Du Bocage Santos-Pinto, Everton Falcão de Oliveira.

**Data curation:** Márcio José de Medeiros.

**Formal analysis:** Ana Isabel do Nascimento, Lisany Krug Mareto, Cláudia Du Bocage Santos-Pinto, Everton Falcão de Oliveira.

**Funding acquisition:** Cláudia Du Bocage Santos-Pinto, Everton Falcão de Oliveira.

**Investigation:** Ana Isabel do Nascimento, Danilo dos Santos Conrado, Lisany Krug Mareto, Micael Viana de Azevedo, João Cesar Pereira da Cunha, Gabriel Serrano Ramires Koch, Laysa Gomes Osório, Samara Tessari Pires, Letícia Suemi Arakaki, Sara Raquel Pinto Borges, Robson Franca Gomes e Silva, Rodrigo Mayer Pucci, João Guilherme de Novaes Corrêa, João Vitor Barrio, Maria Eduarda de Souza Rodrigues, Artur Jorge Bianchi, Márcio José de Medeiros, Maria Elizabeth Araújo Ajalla, Cláudia Du Bocage Santos-Pinto, Everton Falcão de Oliveira.

**Methodology:** Ana Isabel do Nascimento, Micael Viana de Azevedo, João Cesar Pereira da Cunha, Gabriel Serrano Ramires Koch, Laysa Gomes Osório, Samara Tessari Pires, Letícia Suemi Arakaki, Sara Raquel Pinto Borges, Robson Franca Gomes e Silva, Rodrigo Mayer Pucci, João Guilherme de Novaes Corrêa, João Vitor Barrio, Maria Eduarda de Souza Rodrigues, Artur Jorge Bianchi, Márcio José de Medeiros, Ana Paula Sayuri Sato, Maria Elizabeth Araújo Ajalla, Cláudia Du Bocage Santos-Pinto, Everton Falcão de Oliveira.

**Project administration:** Ana Isabel do Nascimento, Maria Elizabeth Araújo Ajalla, Cláudia Du Bocage Santos-Pinto, Everton Falcão de Oliveira.

**Supervision:** Ana Isabel do Nascimento, Maria Elizabeth Araújo Ajalla, Cláudia Du Bocage Santos-Pinto, Everton Falcão de Oliveira.

**Validation:** Ana Isabel do Nascimento, Cláudia Du Bocage Santos-Pinto, Everton Falcão de Oliveira.

**Visualization:** Ana Isabel do Nascimento, Maria Elizabeth Araújo Ajalla, Cláudia Du Bocage Santos-Pinto, Everton Falcão de Oliveira.

**Writing – original draft:** Ana Isabel do Nascimento, Cláudia Du Bocage Santos-Pinto, Everton Falcão de Oliveira.

**Writing – review & editing:** Ana Isabel do Nascimento, Danilo dos Santos Conrado, Lisany Krug Mareto, Micael Viana de Azevedo, João Cesar Pereira da Cunha, Gabriel Serrano Ramires Koch, Laysa Gomes Osório, Samara Tessari Pires, Letícia Suemi Arakaki, Sara Raquel Pinto Borges, Robson Franca Gomes e Silva, Rodrigo Mayer Pucci, João Guilherme de Novaes Corrêa, João Vitor Barrio, Maria Eduarda de Souza Rodrigues, Márcio José de Medeiros, Ana Paula Sayuri Sato, Maria Elizabeth Araújo Ajalla, Cláudia Du Bocage Santos-Pinto, Everton Falcão de Oliveira.

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
