## [Decision Letter · Decision Letter 0]

11 Dec 2025

PONE-D-25-13866Age and Vaccine Information Sources Drive Vaccine Hesitancy: A Household Survey in Central-Western BrazilPLOS One

Dear Dr. Falcão de Oliveira,

Thank you for submitting your manuscript to PLOS ONE. After careful consideration, we feel that it has merit but does not fully meet PLOS ONE’s publication criteria as it currently stands. Therefore, we invite you to submit a revised version of the manuscript that addresses the points raised during the review process.

The reviewers provided valuable and consistent feedback, highlighting the relevance of the topic and the potential contribution of this manuscript to the literature on vaccine hesitancy in Brazil and the Global South. The paper would benefit from further methodological clarification, conceptual precision, and a more parsimonious statistical presentation.

In revising the manuscript, the authors should include a clear justification for the sampling calculations, preferably in an appendix, detailing the assumptions and parameters employed. The rationale for selecting Campo Grande as the study site must also be expanded, explaining how this case contributes to understanding the broader research problem, particularly given its geographical, socioeconomic, and informational specificities. The authors are further encouraged to provide a more detailed description of the cultural adaptation process of the SAGE Work Group Questionnaire for the Brazilian context, ensuring the validity and comparability of the instrument. The operational definitions of hesitant participants (HP) and non-hesitant participants (NHP) should be clarified to reinforce analytical transparency. In the statistical section, a more concise and parsimonious presentation of the bivariate models is recommended, with non-essential details moved to supplementary materials. Throughout the discussion, speculative formulations such as “may have” or “maybe” should be avoided to maintain a scientific tone and restrict interpretations to what the evidence allows to be demonstrated. Finally, the reference to the “neoliberal system” introduced in the discussion requires conceptual elaboration, since the term is not developed elsewhere in the manuscript. Defining it succinctly and connecting it to the behavioral and structural determinants of vaccine hesitancy will enhance theoretical coherence. Overall, the manuscript is promising and well-grounded empirically. Addressing these points will strengthen the methodological robustness, analytical precision, and international relevance of the journal, aligning it more closely with PLOS ONE’s standards.

We look forward to receiving your revised manuscript.

Kind regards,

Ivan Filipe de Almeida Lopes Fernandes, Ph.D.

Academic Editor

PLOS ONE

Journal Requirements:

I have read the journal’s policy and the authors of this manuscript have the following competing interests: EFO is an Academic Editor of PLOS ONE.

6. We note that Figure 1 in your submission contain map images which may be copyrighted. All PLOS content is published under the Creative Commons Attribution License (CC BY 4.0), which means that the manuscript, images, and Supporting Information files will be freely available online, and any third party is permitted to access, download, copy, distribute, and use these materials in any way, even commercially, with proper attribution. For these reasons, we cannot publish previously copyrighted maps or satellite images created using proprietary data, such as Google software (Google Maps, Street View, and Earth). For more information, see our copyright guidelines: http://journals.plos.org/plosone/s/licenses-and-copyright

Additional Editor Comments:

The reviewers provided valuable and consistent feedback, highlighting the relevance of the topic and the potential contribution of this manuscript to the literature on vaccine hesitancy in Brazil and the Global South. The paper would benefit from further methodological clarification, conceptual precision, and a more parsimonious statistical presentation.

In revising the manuscript, the authors should include a clear justification for the sampling calculations, preferably in an appendix, detailing the assumptions and parameters employed. The rationale for selecting Campo Grande as the study site must also be expanded, explaining how this case contributes to understanding the broader research problem, particularly given its geographical, socioeconomic, and informational specificities. The authors are further encouraged to provide a more detailed description of the cultural adaptation process of the SAGE Work Group Questionnaire for the Brazilian context, ensuring the validity and comparability of the instrument. The operational definitions of hesitant participants (HP) and non-hesitant participants (NHP) should be clarified to reinforce analytical transparency. In the statistical section, a more concise and parsimonious presentation of the bivariate models is recommended, with non-essential details moved to supplementary materials. Throughout the discussion, speculative formulations such as “may have” or “maybe” should be avoided to maintain a scientific tone and restrict interpretations to what the evidence allows to be demonstrated. Finally, the reference to the “neoliberal system” introduced in the discussion requires conceptual elaboration, since the term is not developed elsewhere in the manuscript. Defining it succinctly and connecting it to the behavioral and structural determinants of vaccine hesitancy will enhance theoretical coherence. Overall, the manuscript is promising and well-grounded empirically. Addressing these points will strengthen the methodological robustness, analytical precision, and international relevance of the journal, aligning it more closely with PLOS ONE’s standards.

Reviewers' comments:

Reviewer's Responses to Questions

**Comments to the Author**

1. Is the manuscript technically sound, and do the data support the conclusions?

Reviewer #1: Yes

Reviewer #2: Yes

Reviewer #3: Partly

2. Has the statistical analysis been performed appropriately and rigorously? 

Reviewer #1: Yes

Reviewer #2: Yes

Reviewer #3: Yes

3. Have the authors made all data underlying the findings in their manuscript fully available?

Reviewer #1: No

Reviewer #2: Yes

Reviewer #3: No

4. Is the manuscript presented in an intelligible fashion and written in standard English?

Reviewer #1: Yes

Reviewer #2: Yes

Reviewer #3: Yes

5. Review Comments to the Author

Reviewer #1:

Dear Editor,

I consider the topic relevant and current, with important repercussions for discussions regarding vaccine hesitancy, non-vaccination, and refusal. Furthermore, it presents a careful critique of the shortcomings in access to vaccines through health services. The text is quite thought-provoking, simple, and harmonious, motivating reading without causing fatigue or difficulty in interpretation. The study aims to measure vaccine hesitancy and its associated factors among residents of Campo Grande, the capital of the state of Mato Grosso do Sul, Brazil.

This is a very important topic for current discussions regarding vaccination coverage, which has been declining in the country, where the Immunization Program is powerful and recognized as one of the best in the world. The authors very well justify the importance of conducting the study and, of course, publishing the results, which makes it reasonable to consider its publication in a journal like PlosOne as timely.

Regarding materials and methods

The authors state that this is a cross-sectional study, aligned with a household survey conducted in Campo Grande to estimate vaccination coverage in the municipality, with data collected between September 2022 and October 2023.

In this section, I initially had doubts regarding the methodology; it wasn't entirely clear that this article is part of the results of the household survey (also conducted by them?) used for a different purpose (to estimate vaccination coverage). By stating in the first part of this section that the objective was to estimate vaccination coverage, the initial impression is that there was a change in the objective of the work. I believe a small correction will make the text clearer.

Regarding the study population and data collection, the methodology employed is very well described, allowing for the replication of the study for both sampling and survey application.

The statistical analysis employed is also appropriate for the purpose of the study. The results were presented clearly, and the discussion considered previous studies showing possible comparisons.

Reviewer #2: The article presents a theme of significant importance for public health, as it examines health vulnerability within diverse social contexts. The research displays appropriate scientific rigor and complies with ethical requirements. I endorse its publication.

Reviewer #3: This manuscript addresses a highly relevant and policy-sensitive topic—vaccine hesitancy—in the Brazilian context. The dataset is valuable, and the survey is methodologically sound, offering potential for a meaningful contribution to global public health discussions. However, to meet the standards expected in a high-impact international journal such as PLOS ONE, the paper requires major revisions in several areas of structure, contextualization, and theoretical framing.

1. Literature Review and Theoretical Framework

The literature review is currently too narrow and must be substantially expanded.

While the 3C and 5C models are important conceptual tools for understanding vaccine hesitancy, the manuscript should also engage with other established frameworks in the health and behavioral sciences.

Classic public health literature has long discussed the trade-off between self-protection and vaccination, noting that in periods of lower perceived risk, individuals tend to vaccinate less—a phenomenon well documented before the COVID-19 era.

The authors should also incorporate literature on risk perception, behavioral economics, and health communication, as well as the political economy of vaccination in contexts of misinformation and institutional fragmentation. The Brazilian case is particularly relevant here, as federal authorities—including the former president—publicly questioned vaccination and containment measures, contributing to public confusion. This political dimension must be reflected and supported with appropriate scholarly references.

2. Contextualization of the Brazilian Case

Brazil has one of the world’s most longstanding and successful immunization programs, established in the 1970s during the military regime, which achieved major public health milestones through strong coordination, communication, and outreach.

The authors should discuss:

The historical trajectory of the National Immunization Program (PNI) and its role in shaping public trust.

The recent decline in vaccination coverage since 2016, linked to fiscal constraints and reduced federal investment in vaccination campaigns.

The loss of public health communication capacity—propaganda, media campaigns, and health education—which used to sustain confidence in vaccines.

Campo Grande should also be contextualized geographically and socioeconomically. How representative is this municipality within Brazil’s continental diversity? Is it a border region influenced by migration, poverty, or digital misinformation networks?

3. Data Presentation and Statistical Rigor

The paper presents interesting results, but it lacks sufficient statistical detail and transparency.

The authors should:

Include appendices or supplementary materials with the full survey instrument, descriptive tables, and regression outputs.

Explore heterogeneity across age groups, given that older generations experienced robust vaccination programs, while younger individuals were socialized in an era of digital disinformation.

Discuss how social media usage patterns and regional political contexts might interact with hesitancy. Including a map or stratification by clusters could strengthen the empirical contribution.

4. Discussion and Policy Implications

The discussion needs to move beyond description. The authors should draw clearer policy lessons from their findings—how public health systems can rebuild confidence, counter misinformation, and restore vaccination coverage.

Comparative insights from other Latin American or Global South contexts could make the paper more internationally relevant. The authors could also explore policy alternatives, such as communication strategies, digital literacy programs, and targeted outreach to younger populations.

5. Overall Assessment

The study is promising and empirically rich, but its current framing makes it read more as a localized case study than as a paper contributing to global public health theory.

With a deeper theoretical engagement, a stronger connection to Brazil’s historical and political context, and a clearer presentation of data, the manuscript has real potential to make an important contribution to the literature on vaccine hesitancy, health communication, and pandemic governance.

6. PLOS authors have the option to publish the peer review history of their article (what does this mean?). If published, this will include your full peer review and any attached files.

Reviewer #1: No

Reviewer #2: No

Reviewer #3: No

---

## [Author Response · Author response to Decision Letter 1]

4 Feb 2026

Editor’s comments

In revising the manuscript, the authors should include a clear justification for the sampling calculations, preferably in an appendix, detailing the assumptions and parameters employed. The rationale for selecting Campo Grande as the study site must also be expanded, explaining how this case contributes to understanding the broader research problem, particularly given its geographical, socioeconomic, and informational specificities. The authors are further encouraged to provide a more detailed description of the cultural adaptation process of the SAGE Work Group Questionnaire for the Brazilian context, ensuring the validity and comparability of the instrument. The operational definitions of hesitant participants (HP) and non-hesitant participants (NHP) should be clarified to reinforce analytical transparency. In the statistical section, a more concise and parsimonious presentation of the bivariate models is recommended, with non-essential details moved to supplementary materials. Throughout the discussion, speculative formulations such as “may have” or “maybe” should be avoided to maintain a scientific tone and restrict interpretations to what the evidence allows to be demonstrated. Finally, the reference to the “neoliberal system” introduced in the discussion requires conceptual elaboration, since the term is not developed elsewhere in the manuscript. Defining it succinctly and connecting it to the behavioral and structural determinants of vaccine hesitancy will enhance theoretical coherence. Overall, the manuscript is promising and well-grounded empirically. Addressing these points will strengthen the methodological robustness, analytical precision, and international relevance of the journal, aligning it more closely with PLOS ONE’s standards.

Author response: We thank the Editor for their comments and suggestions, which substantially enhanced the overall quality of the manuscript. All recommendations were fully addressed. An S1 Appendix detailing the sampling justification and methodology was uploaded alongside the data questionnaire. Nevertheless, we maintained a cautious use of language indicating possibility, given the methodological design of our work, which does not allow causal inferences beyond hypothesis generation for future research. In addition, a definition for the neoliberal system was included in the manuscript (page 18, lines 378 - 382)

Reviewer #1

Regarding materials and methods

The authors state that this is a cross-sectional study, aligned with a household survey conducted in Campo Grande to estimate vaccination coverage in the municipality, with data collected between September 2022 and October 2023.

In this section, I initially had doubts regarding the methodology; it wasn't entirely clear that this article is part of the results of the household survey (also conducted by them?) used for a different purpose (to estimate vaccination coverage). By stating in the first part of this section that the objective was to estimate vaccination coverage, the initial impression is that there was a change in the objective of the work. I believe a small correction will make the text clearer.

Author response: Thank you for this observation. We acknowledge that the original wording may have lacked clarity and could have led to confusion regarding the methodology. We have revised the paragraph to clearly describe the household survey design and its alignment with the objective of assessing vaccine hesitancy. The revised text now reads as follows: “This cross-sectional study used data collected through a household survey conducted in Campo Grande, Mato Grosso do Sul, Brazil, between September 2022 and October 2023, with the primary objective of estimating vaccination coverage in the municipality. For the purposes of this analysis, data on vaccine hesitancy were also collected and analyzed”.

Regarding the study population and data collection, the methodology employed is very well described, allowing for the replication of the study for both sampling and survey application.

The statistical analysis employed is also appropriate for the purpose of the study. The results were presented clearly, and the discussion considered previous studies showing possible comparisons.

Author response: We thank Reviewer #1 for their constructive comments and careful review of the manuscript.

Reviewer #2

The article presents a theme of significant importance for public health, as it examines health vulnerability within diverse social contexts. The research displays appropriate scientific rigor and complies with ethical requirements. I endorse its publication.

Author response: We thank Reviewer #2 for their constructive comments and careful review of the manuscript.

Reviewer #3

1. Literature Review and Theoretical Framework

The literature review is currently too narrow and must be substantially expanded. While the 3C and 5C models are important conceptual tools for understanding vaccine hesitancy, the manuscript should also engage with other established frameworks in the health and behavioral sciences. Classic public health literature has long discussed the trade-off between self-protection and vaccination, noting that in periods of lower perceived risk, individuals tend to vaccinate less—a phenomenon well documented before the COVID-19 era.

Author response: We thank the reviewer for this comment and acknowledge the importance of broader theoretical frameworks in the study of vaccine hesitancy. Nevertheless, the manuscript intentionally focuses on the 3C model, as this framework directly underpins the WHO-recommended instrument used for data collection and analysis. While the 5C model is conceptually relevant, it was proposed after the development of the questionnaire applied in this study and is therefore addressed only in the discussion where appropriate. This approach was adopted to preserve theoretical consistency and maintain focus on the study’s central research question.

To address the reviewer’s comment, we have added a paragraph to the Discussion section addressing the literature on risk perception (page 18, lines 389 – 396).

The authors should also incorporate literature on risk perception, behavioral economics, and health communication, as well as the political economy of vaccination in contexts of misinformation and institutional fragmentation. The Brazilian case is particularly relevant here, as federal authorities—including the former president—publicly questioned vaccination and containment measures, contributing to public confusion. This political dimension must be reflected and supported with appropriate scholarly references.

Author response: We agree and thank the reviewer for highlighting the relevance of risk perception, health communication, and political-institutional contexts in shaping vaccine-related attitudes. In response to this comment, we have added a paragraph to the Discussion section situating our findings within the literature on risk perception and health communication, drawing on the WHO SAGE framework and international evidence on vaccine confidence (page 20, lines 430 – 445). This includes references to how political and institutional contexts may influence public trust, as documented in comparative studies across high- and middle-income countries. Given the cross-sectional design and the scope of the WHO-recommended survey instrument used in this study, these aspects are discussed as contextual factors rather than primary analytical determinants, in order to avoid causal interpretations not supported by the data.

2. Contextualization of the Brazilian Case

Brazil has one of the world’s most longstanding and successful immunization programs, established in the 1970s during the military regime, which achieved major public health milestones through strong coordination, communication, and outreach. The authors should discuss:

The historical trajectory of the National Immunization Program (PNI) and its role in shaping public trust.

The recent decline in vaccination coverage since 2016, linked to fiscal constraints and reduced federal investment in vaccination campaigns.

The loss of public health communication capacity—propaganda, media campaigns, and health education—which used to sustain confidence in vaccines. Campo Grande should also be contextualized geographically and socioeconomically. How representative is this municipality within Brazil’s continental diversity? Is it a border region influenced by migration, poverty, or digital misinformation networks?

Author response: We included a brief description of the development of the Brazilian Immunization Program (page 3, lines 44 – 62), and also described important characteristics of the municipality of Campo Grande in the Methods section (page 5, lines 105 – 115).

3. Data Presentation and Statistical Rigor

The paper presents interesting results, but it lacks sufficient statistical detail and transparency. The authors should:

Include appendices or supplementary materials with the full survey instrument, descriptive tables, and regression outputs.

Explore heterogeneity across age groups, given that older generations experienced robust vaccination programs, while younger individuals were socialized in an era of digital disinformation.

Discuss how social media usage patterns and regional political contexts might interact with hesitancy. Including a map or stratification by clusters could strengthen the empirical contribution.

Author response: We appreciate the feedback and considered all of it during the manuscript review. S2 and S3 Appendices detailing the methodology, including the data analysis, were included in the revised version of the manuscript.

Regarding heterogeneity across age groups, our data already support a generational interpretation, as hesitant participants were, on average, younger than non-hesitant individuals (mean = 43.8 years [SD = 16.3] vs. mean = 49.6 years [SD =18.0]). This pattern suggests an age gradient consistent with generational differences rather than an abrupt threshold effect, supporting the use of age as a continuous variable in the main analysis while allowing for interpretative discussion of generational dynamics. To address the reviewer’s comment and explore heterogeneity across age groups, we conducted additional descriptive analyses stratified by age categories, while retaining age as a continuous variable in the main regression model to preserve statistical power and avoid arbitrary cutoffs.

4. Discussion and Policy Implications

The discussion needs to move beyond description. The authors should draw clearer policy lessons from their findings—how public health systems can rebuild confidence, counter misinformation, and restore vaccination coverage. Comparative insights from other Latin American or Global South contexts could make the paper more internationally relevant. The authors could also explore policy alternatives, such as communication strategies, digital literacy programs, and targeted outreach to younger populations.

Author response: We appreciate the reviewer’s suggestion and have incorporated a paragraph aimed at contributing to public health policies. We also included a brief paragraph regarding the lack of studies developed on general vaccination and discussed our vaccine hesitancy percentage compared to other LATAM studies.

5. Overall Assessment

The study is promising and empirically rich, but its current framing makes it read more as a localized case study than as a paper contributing to global public health theory. With a deeper theoretical engagement, a stronger connection to Brazil’s historical and political context, and a clearer presentation of data, the manuscript has real potential to make an important contribution to the literature on vaccine hesitancy, health communication, and pandemic governance.

Author response: We thank the thoughtful analysis of our work, and hopefully, the changes applied will highlight the importance of publications regarding vaccine hesitancy

---

## [Decision Letter · Decision Letter 1]

15 Apr 2026

Age and Vaccine Information Sources Drive Vaccine Hesitancy: A Household Survey in Central-Western Brazil

PONE-D-25-13866R1

Dear Dr. Falcão de Oliveira,

We’re pleased to inform you that your manuscript has been judged scientifically suitable for publication and will be formally accepted for publication once it meets all outstanding technical requirements.

Kind regards,

Ivan Filipe de Almeida Lopes Fernandes, Ph.D.

Academic Editor

PLOS One

Additional Editor Comments (optional):

Reviewers' comments:

Reviewer's Responses to Questions

**Comments to the Author**

1. If the authors have adequately addressed your comments raised in a previous round of review and you feel that this manuscript is now acceptable for publication, you may indicate that here to bypass the “Comments to the Author” section, enter your conflict of interest statement in the “Confidential to Editor” section, and submit your "Accept" recommendation.

Reviewer #3: All comments have been addressed

2. Is the manuscript technically sound, and do the data support the conclusions?

Reviewer #3: Yes

3. Has the statistical analysis been performed appropriately and rigorously? 

Reviewer #3: Yes

4. Have the authors made all data underlying the findings in their manuscript fully available?

Reviewer #3: Yes

5. Is the manuscript presented in an intelligible fashion and written in standard English?

Reviewer #3: Yes

6. Review Comments to the Author

Reviewer #3: I am satisfied. TThe authors addressed all comments. Congratularions. I hope to see the paper soon published.

7. PLOS authors have the option to publish the peer review history of their article (what does this mean?). If published, this will include your full peer review and any attached files.

Reviewer #3: **Yes:**GUSTAVO ANDREY DE ALMEIDA LOPES FERNANDES

---

## [Editor Report · Acceptance letter]

PONE-D-25-13866R1

PLOS One

Dear Dr. Falcão de Oliveira,

I'm pleased to inform you that your manuscript has been deemed suitable for publication in PLOS One. Congratulations! Your manuscript is now being handed over to our production team.

Kind regards,

on behalf of

Dr. Ivan Filipe de Almeida Lopes Fernandes

Academic Editor

PLOS One